# Rapid and Sensitive Detection of Bisphenol A Based on Self-Assembly

**DOI:** 10.3390/mi11010041

**Published:** 2019-12-30

**Authors:** Haoyue Luo, Xiaogang Lin, Zhijia Peng, Min Song, Lifeng Jin

**Affiliations:** Key Laboratory of Optoelectronic Technology and Systems of Ministry of Education of China, Chongqing University, Chongqing 400044, China; 20133029@cqu.edu.cn (H.L.); 201808021026@cqu.edu.cn (Z.P.); 201908131078@cqu.edu.cn (M.S.); 20152394@cqu.edu.cn (L.J.)

**Keywords:** alternating current (AC) electrokinetics, bisphenol A, self-assembly, biosensor

## Abstract

Bisphenol A (BPA) is an endocrine disruptor that may lead to reproductive disorder, heart disease, and diabetes. Infants and young children are likely to be vulnerable to the effects of BPA. At present, the detection methods of BPA are complicated to operate and require expensive instruments. Therefore, it is quite vital to develop a simple, rapid, and highly sensitive method to detect BPA in different samples. In this study, we have designed a rapid and highly sensitive biosensor based on an effective self-assembled monolayer (SAM) and alternating current (AC) electrokinetics capacitive sensing method, which successfully detected BPA at nanomolar levels with only one minute. The developed biosensor demonstrates a detection of BPA ranging from 0.028 μg/mL to 280 μg/mL with a limit of detection (LOD) down to 0.028 μg/mL in the samples. The developed biosensor exhibited great potential as a portable BPA biosensor, and further development of this biosensor may also be useful in the detection of other small biochemical molecules.

## 1. Introduction

Bisphenol A (BPA) is an important organic chemical raw material, which is widely used in the production of plastic products and fire retardant. BPA is an endocrine disruptor, which can mimic human hormones and maybe lead to negative health effects [1,2]. Studies have shown that BPA can contact humans through the skin, respiratory tract, digestive tract, and other channels. After BPA enters the body, it combines with intracellular estrogen receptors and produces estrogenic or anti-estrogenic effects through a variety of reaction mechanisms, thereby affecting endocrine, reproductive, and nervous systems, as well as causing cancer and other adverse effects [3,4,5]. Therefore, a rapid and sensitive detection method of BPA is of great significance.

At present, the detection methods of BPA mainly include liquid chromatography with an electrochemical method [6], chromatography-mass spectrometry [7,8], surface-enhanced Raman scattering [9], enzyme-linked immunosorbent assay (ELISA) [10], etc. These methods are complicated and require expensive equipment with professional personnel. The movement of biomolecular molecules in the detection process relies on natural diffusion, which is time-consuming and cannot meet the requirements for the rapid detection of BPA. For example, Pasquale et al. [11] have proposed a method to determine BPA levels in fruit juices by liquid chromatography coupled to tandem mass spectrometry. However, it cost about 15 min to accomplish the detection of BPA. Sheng et al. [12] have developed an optical biosensor based on fluorescence, but they required the addition of labels to generate the sensor response. Xue et al. [13] have reported a novel SPR biosensor that combines a binding inhibition assay with functionalized gold nanoparticles to allow for the detection of trace concentrations of BPA. However, the biosensor detected BPA in 60 min, which was too long for rapid detection. Recently, various types of electrochemical sensors have attracted more attention. These electrochemical sensors are always modified via different sensing materials, such as molecularly imprinted polymers [14], carbon nanotube [15,16], graphene [17,18], nanocomposites [19,20], and metal composites [21]. For example, Maryam et al. [22] have used an electroactive label-based aptamer to detect bisphenol A in serum samples with the linear range of 0.2–2 nM. Lee et al. [23] have reported a simple and label-free colorimetric aptasensor for BPA detection. They have used the spectroscopic methods, which was time-consuming and could not achieve point-of-care testing. Inroga et al. [24] have used gold leaf-like microstructures to develop a tyrosinase-based biosensor for bisphenol A detection with the limit of detect of 0.077 μmol/L. Liu et al. [25] have developed an electrochemical enzyme biosensor bearing biochar nanoparticles as signal enhancers for bisphenol A detection in water. Peng et al. [26] have developed a signal-enhanced lateral flow strip biosensor for the visual and quantitative detection of BPA based on the poly amidoamine (PAMAM) binding with antibody. The detection would cost 10 min. Liu et al. [27] have developed a solution-gated graphene transistor (SGGT) modified with DNA molecules in a microfluidic system for a recycling detection of BPA. Compared with most of the recent publications, which have reported the electrochemical method to achieve the detection of BPA, the results of this work may not be better than them. However, in this work, we have demonstrated a novel method to detect BPA based on self-assembly technology and alternating current (AC) electrokinetics effect. To our knowledge, there has not been any report that detects BPA with self-assembly technology and an AC electrokinetics effect. So, it has the tremendous potential to detect BPA more sensitively with a lower limit of detection in the near future.

In this work, a rapid, highly sensitive BPA biosensor based on self-assembly technology and AC electrokinetics (ACEK) method was developed, which could accomplish the rapid and sensitive detection of BPA. The surface of the interdigital electrode was functionalized with the antibody of BPA via self-assembly and used for the specific capture of antigen of BPA. ACEK is used to realize the enrichment of biomolecules through the manipulation of microfluidic and nanoparticles by an AC electric field [28,29]. Compared with traditional detection methods, the ACEK method has the advantages of less sample consumption, faster detection, and a simpler procedure with an appropriate AC signal applied to microelectrode sensors in sample fluids. Then, ACEK microflows accelerate molecules of BPA binding [30,31]. With the enrichment of the target molecules, the interfacial capacitance of the interdigital electrode will change. Therefore, the change of the interfacial capacitance can be used to characterize the concentration of sample solution. Compared with the previous work [32], there are some differences in this work. Firstly, in this work, we used the BPA antibody to achieve the detection of BPA, while an aptamer probe was used to detect BPA in the previous work. Secondly, we used the self-assembly technology to form a self-assembled monolayer with the mercaptan containing alkyl chains immobilized on the gold electrodes via Au–S bonds. Then, N-hydroxysuccinimide (NHS) and 1-(3-dimethylaminopropyl)-3-ethyl carbon diimide hydrochloride (EDC) were used as the crosslinking agent to assist in the formation of amide bonds between the carboxyl group of the self-assembled monolayer and the amino group of the BPA antibody. With these processes, the BPA antibody was immobilized onto the electrodes stably, while a BPA aptamer specific to BPA with the 5′ thiol modifier C6 SH (Fisher Scientific, PA) was directly immobilized onto the electrodes in the previous work. Thirdly, in this work, the interdigital microelectrodes had interdigitated arrays with widths of 10 μm separated by 10-μm gaps. However, the interdigital microelectrodes had interdigitated arrays with widths of 6 μm separated by 6-μm gaps in the previous work. Lastly, in this work, a 10 kHz AC signal was applied to the microelectrodes, but a 20 kHz AC signal was applied to the microelectrodes in the previous work.

With the improved biosensor design and use of an antigen and antibody, we successfully detected BPA at the nanomolar level within only one minute. Moreover, the developed BPA biosensor is expected to be used in the detection of biological fluids. It is of great significance to promote the detection of other small biochemical molecules.

## 2. Materials and Methods

### 2.1. Materials and Reagents

The buffer solution 10 × PBS was purchased from Solarbio (Beijing, China). The 10 × PBS was diluted in deionized water to make the working solution at 0.05 × PBS, 0.01 × PBS for diluting other substances. At the same time, the 0.01 × PBS was used as the background solution. 11-mercaptoundecanoic acid (MUA) was purchased from Yuanye Bio-Technology Co., Ltd., (Shanghai, China) The MUA was dissolved in anhydrous ethanol to prepare 5 mmol/L of MUA solution for forming the gold–sulfur bond. 1-(3-dimethylaminopropyl)-3-ethyl carbon diimide hydrochloride (EDC) was obained from Sigma (St. Louis, MI, USA); EDC was dissolved in 0.05 × PBS to make 0.4 mol/L EDC solution. N-hydroxysuccinimide (NHS) was acquired from biotopped (Beijing, China), and NHS was dissolved in 0.05 × PBS to prepare 0.1 mol/L NHS solution. Then, we mixed the two solution 1 to 4. Ethanolamine was purchased from MACKLIN (Shanghai Macklin Biochemical, Ltd., Shanghai, China). Ethanolamine was diluted with 0.05 × PBS to make 1 mol/L solution for closing. The BPA antibody and antigen were purchased from QF Biosciences Co., Ltd., (Shanghai, China). The antibody was diluted with 0.05 × PBS to make two concentrations of 5.3 μg/L and 5.3 μg/mL, and the antigen was diluted in 0.01 × PBS to make the solution that included 0.028 μg/mL, 0.28 μg/mL, 2.8 μg/mL, 28 μg/mL, and 280 μg/mL for testing. Antigen and antibody ware stored at −20 °C. In order to guarantee the accuracy of detection, preparation of the solution was carried out on a clean bench.

### 2.2. Preparation of Interdigital Microelectrodes

In this work, the interdigital microelectrodes were fabricated on silicon wafers, which had interdigitated arrays with widths of 10 μm separated by 10-μm gaps [32]. Before the detection, the interdigital microelectrodes should be modified with the following steps, as shown in Figure 1. Firstly, they are immersed in acetone for 4 min with ultrasonic cleaning, rinsed in absolute ethyl alcohol for 3 min with ultrasonic cleaning, rinsed in deionized water for 3 min with ultrasonic cleaning, and dried with a drying oven. Secondly, MUA was added to the surface of the gold electrodes for forming the Au–S bonds and realizing the self-assembled monolayer [33,34]. Then, sensors were placed in the incubator overnight with the temperature at 25 °C. Thirdly, before the EDC and NHS solution were added to the electrodes, the electrodes surface should be cleaned with absolute ethyl alcohol and blow dried with nitrogen; then, the sensors should be placed in an incubator for 2 h. After activation of the carboxyl group, the electrodes’ surface should be cleaned with deionized water and blow dried with nitrogen; then, the chambers were pasted on the sensors. After that, 10 μL of antibody should be added to the surface of the sensors, which are then placed in an incubator for 3 h with the temperature at 37 °C. EDC and NHS were used as the crosslinking agent to assist in the formation of amide bonds between the carboxyl group of the self-assembled monolayer and the amino group of the BPA antibody. With these processes, the BPA antibody was immobilized onto the electrodes stably. Finally, in order to enhance the specificity of detection, ethanolamine was utilized to close the unbound active sites of the electrodes’ surface. Then, the sensors were placed in an incubator for 1 h with the temperature at 25 °C. After the above steps, the modification of the electrodes surface was accomplished. The sensor needs to be cleaned after each sensing procedure; it is possible to wash off the antigen only and reuse the sensor, which indicates that the sensor has an expected reusability of five times before the performance of the sensor is believed to be dissatisfactory.

### 2.3. Apparatus and Methods

An impedance analyzer of model IM3536 (HIOKI, Ueda, Japan) was a high-precision apparatus to detect the impedance, capacitance, and resistance of the modified interdigital electrode after being dropped different concentrations of antigens solution. Firstly, different frequencies (1 kHz, 10 kHz, 20 kHz) and different voltages (100 mV, 600 mV, 1.1 V) were applied to the experiments, respectively. Here, Figure 2 showed the relationship between the normalized capacitance and different concentrations of BPA with a 10-kHz AC signal of different voltages (100 mV, 600 mV, and 1.1 V) applied to the interdigital electrodes. Finally, a 10-kHz AC signal of 600 mV was selected to be applied to the interdigital electrodes via the impedance analyzer of model IM3536 as the measuring signal. At the same time, the voltage used in the experiments is rms. In this work, before detecting antigen concentrations, each sensor should detect the background solution. However, the measurements of the background solution are made in this work only and do not need to be used in commercial application. We detected the background solution as the blank control group to highlight the antibody–antigen binding response to the change in capacitance of electrodes. Then, 10 μL of different concentrations of antigens solution were added to interdigital electrodes. The impedance, capacitance, and resistance of the modified interdigital electrodes were detected within one minute. The normalized capacitance change rate of the sensor was computed to demonstrate antigen–antibody binding, which was shown with the slope of normalized capacitance versus time (%/min). Then, the slope was linearly fitted by the least square method. The normalized capacitance was computed as *C_t_*/*C*_0_, where *C_t_* is the capacitance value at time *t* and *C*_0_ is the capacitance value at time zero [32].

### 2.4. Electrical Double Layers

What happens when a solution comes into contact with a solid metal surface? Helmholtz built a model to attempt to explore this question [35]. He called it model H as the left half of the dotted line in Figure 3. As is shown in Figure 3, this model can be equivalent to a flat plate capacitor, and the relationship between the charge density (*σ*) on one side and the potential (*V*) difference between the two layers is described by the following equations [35,36], where *d* is the distance of center of the positive and negative charge.
(1)$$\sigma =\frac{\varepsilon \varepsilon_{0}}{d}V$$
(2)$$\frac{\partial \sigma }{\partial V}=C_{H}=\frac{\varepsilon \varepsilon_{0}}{d}$$

We can conclude the capacitance ($C_{H}$) of the flat plate capacitor from Equation (1). Hence, the H model can successfully describe the common electrochemical phenomenon with two basic equations. However, the Helmholtz layer shows an obvious defect, as shown in Figure 3. In the corollary of the equation, $C_{H}$ is a constant, but in the experiment, there are spreading layers, leading to $C_{H}$ not accurately describing the surface change. We describe the diffuse layer as $C_{D}$ and call the electrical double layer (EDL) as $C_{d}$ [36], which will be affected by the relative potential and electrolyte concentration. From this, we can deduce that $C_{d}$ is equal to $C_{H}$ in series with $C_{D}$. The value of $C_{d}$ can be determined by the following equation.
(3)$$C_{d}=\frac{C_{H}C_{D}}{C_{H}+C_{D}}$$

As mentioned above, both the charging and discharging processes of the electrical double layer (EDL) are similar to those of the parallel plate. When the electrolyte is added to the surface of interdigitated microelectrodes, the electrolyte will contact with the surface of the microelectrode in Figure 3. Thus, it can be equivalent to the parallel plate, as shown in Figure 4 [37]. When the electrode is bare, the interface capacitance of it can be described by the following equation.
(4)$$C_{int,0}=\frac{A_{int}}{\frac{\lambda_{d}}{\varepsilon_{s}}}$$
where $\varepsilon_{s}$ is the permittivity of the solution, $A_{int}$ is the electrode area, and $\lambda_{d}$ is the electrical double layer (EDL) thickness.

When antibodies are immobilized to the surface of microelectrodes via self-assembly, the interfacial capacitance $C_{int}$ is expected to change to
(5)$$C_{int,ab}=\frac{A_{b}}{\frac{\lambda_{d}}{\varepsilon_{s}}+\frac{d_{ab}}{\varepsilon_{t}}}$$
where $\varepsilon_{t}$ is the permittivity of the antibody, $A_{b}$ is the effective area after antibodies are immobilized to the surface of microelectrodes, and $d_{ab}$ is the antibody thickness.

In the experiments, the solution including the antigens was added to the surface of interdigitated microelectrodes, on which the antibodies were immobilized. When the antigens bind to antibodies, the molecular deposition on the sensor surface become thicker, and the interfacial capacitance $C_{int,ab}$ will be expressed with
(6)$$C_{int,ag}=\frac{A_{g}}{\frac{\lambda_{d}}{\varepsilon_{s}}+\frac{d_{ab}}{\varepsilon_{t}}+\frac{d_{ag}}{\varepsilon_{p}}}$$
where $\varepsilon_{p}$ is the permittivity of the antigen, $d_{ag}$ is the antigen thickness, and $A_{g}$ is the effective area of the interfacial capacitor after the binding of an antigen to an antibody. Assume that the area of interfacial capacitance is equal before and after the binding. From Equation (6), we can see that the diminution of interfacial capacitance can be lead by the thickness increase of the dielectric layer. In this work, the biosensing utilizes the change of interfacial capacitance $C_{int,ab}$. Thus, the relative changes of interfacial capacitance are used to detect the specific binding of antigens to antibodies. The relative changes of interfacial capacitance are
(7)ΔCCint,ab=Cint,ab−Cint,agCint,ab=−dag/(εpεsλd+εpεtdab+dag).

Consequently, the value of $\Delta C/C_{int,ab}$ can be used to detect the biomolecular interactions. Beyond that, measuring the value of $\Delta C/C_{int,ab}$ can overcome the experimental difference caused by the different surface roughness of each electrode and the difference of experimental treatment in each group, and improve the accuracy of experimental results.

### 2.5. The Binding Mechanism of Interdigital Electrode Surface

In this work, we adopt the interdigital electrodes as the sensors. The interdigital microelectrodes are finger-shaped in its surface. At the same time, this shape can be utilized to achieve the effect of ACEK. On the surface of the self-assembled electrode, there are two forms of binding antigens to antibodies.

In Figure 5, when the AC signal is not applied to the interdigital electrodes, the antigens binding to antibodies only depend on the deposition. In this case, only a small fraction of antigens have the chance to bind to the antibodies. This process takes a long time and does not guarantee the activity of antigens and antibodies. However, when an AC signal is applied to the interdigital electrodes, the ACEK effect is generated on the surface of the electrodes. Under the action of the ACEK effect, more and more antigens are rapidly enriched near the antibodies to promote the binding [38], thus improving the accuracy, rapidity, and sensitivity of detection. Hence, in this work, the ACEK effect was used on the electrodes to accelerate the binding.

## 3. Results and Discussions

### 3.1. Detection of Antigen with 5.3 μg/L Antibody

In this study, different concentrations of BPA standers (1.2 $\times 10^{-7}$ mol/L to 1.2 $\times 10^{-4}$ mol/L) were tested to evaluate the performance of the method. A 10-kHz AC signal of 600 mV was applied to measure capacitance of the biosensor for 60 s.

Figure 6a shows the relationship between the normalized capacitance and different concentrations of BPA. Obviously, the change of capacitance was linear, and the rate of change of the curve increased as the concentrations of BPA increased, corresponding to the degree of binding between antibodies and antigens. The change rate of normalized capacitance curves was found by least square linear fitting, which provided a quantitative index of antibody–antigen binding. In Figure 6a, the slope of these capacitance curves was found to be −10.6‰/min, −14.6‰/min, −24.7‰/min, and −37.7‰/min for BPA levels at 1.2 $\times 10^{-7}$ mol/L, 1.2 $\times 10^{-6}$ mol/L, 1.2 $\times 10^{-5}$ mol/L, and 1.2 $\times 10^{-4}$ mol/L, respectively.

In Figure 6b, we calculated the averages and standard deviations (SDs) of the biosensor response and demonstrated the correlation between the concentration of BPA and change rate of the capacitance. The range of 1.2 $\times 10^{-7}$ mol/L to 1.2 $\times 10^{-4}$ mol/L BPA samples showed change rates of −11.62‰/min ± 2.08‰/min, −18.53‰/min ± 3.93‰/min, −25.07‰/min ± 2.73‰/min, and −30.43‰/min ± 2.56‰/min, respectively. In the range of 1.2 $\times 10^{-7}$ mol/L to 1.2 $\times 10^{-4}$ mol/L BPA, *dC/dt* was logarithmically dependent on the concentration of BPA. A negative linear correlation between *dC/dt* and the concentration of BPA was observed. The dependence is expressed as y(‰/min)= −6.912x + 3.045 with a Pearson correlation coefficient $R^{2}$ = 0.985.

In the experiments, we used the impedance analyzer IM3536 to detect the change of the capacitance of the sensor when antigen–antibody binding. However, the mechanism of measurement of the impedance analyzer IM3536 is to measure the impedance and the phase angle of the device, and the capacitance is calculated from the impedance and phase angle. Considering that the developed sensor is not a pure resistor or pure capacitor, the impedance and phase angle of the sensor will change at any moment when the sensor is immersed into solution, so the capacitance has oscillation. However, the oscillation is in the range of error measurement of the impedance analyzer IM3536. In order to mitigate these oscillations in the capacitance used in the sensor transduction, it’s an effective way to slow down the speed of measurement or take the average of multiple measurements.

### 3.2. Detection of Antigen with 5.3 μg/mL Antibody

To verify the difference in detection between electrodes modified with different antibody concentrations, 5.3 μg/mL of antibody was immobilized on the interdigital electrodes. The experimental conditions are consistent with the experiment above.

Figure 7a displays the change rate of the normalized capacitance, which was found to be −11.5‰/min, −20.0‰/min, −25.5‰/min, and −30.8‰/min for BPA concentrations at 1.2 $\times 10^{-7}$ mol/L, 1.2 $\times 10^{-6}$ mol/L, 1.2 $\times 10^{-5}$ mol/L, and 1.2 $\times 10^{-4}$ mol/L, respectively. The averages and standard deviations (SDs) of the biosensor response were exhibited in Figure 7b. From it, we can evaluate that the concentration of BPA was negatively correlated with *dC/dt*, and the linear correlation is expressed as y(‰/min)= −6.93x + 1.47 with a Pearson correlation coefficient of $R^{2}$ = 0.989. It follows that the electrode with a concentration of 5.3 μg/mL can also detect the BPA levels range from 1.2 $\times 10^{-7}$ mol/L to 1.2 $\times 10^{-4}$ mol/L, and the linearity and correlation are better.

Table 1 lists the quantitative results of the recent publications with different methods to detect BPA. Compared with most of the recent publications, which have reported the electrochemical method to achieve the detection of BPA, the results of this work may not be better than them. However, in this work, we have demonstrated a novel method to detect BPA based on self-assembly technology and the AC electrokinetics effect. There is nearly no report to detect BPA with self-assembly technology and the AC electrokinetics effect. So, it has the tremendous potential to detect BPA more sensitively with a lower limit of detection in the near future.

## 4. Conclusions

In this work, a rapid, highly sensitive BPA biosensor based on self-assembly technology and the AC electrokinetics (ACEK) effect has been proposed. A higher concentration of BPA solution was dropped on the self-assembly interdigital electrode sensor, a larger number of antibody–antigen binding occurred, and a larger normalized capacitance change rate was detected for the sensor. In this work, we used antigen–antibody-specific binding to detect BPA. At the same time, the limit of the biosensor is 1.2 $\times 10^{-7}$ mol/L, which is better than some existing detection methods. For example, Sun et al. [39] have used the oscillopolarogaphic method to detect BPA in food packaging materials with a limit of detection of 4.4 $\times 10^{-6}$ mol/L. Yan et al. [40] have developed a simple and renewable nanoporous gold-based electrochemical sensor for BPA detection with a limit of detection of 4.3 $\times 10^{-7}$ mol/L. The present work could successfully detect BPA at nanomolar (nM) levels, which was higher than the results in the previous work [32], which could successfully detect BPA at femto molar (fM) levels. Although the results of the present work were not better than the results in the previous work, we have used a novel method (self-assembly technology) to achieve the detection of BPA. At the same time, in order to acquire better results, we are improving the conditions, processes, and materials of the experiments. The novel method has the tremendous potential to detect BPA more sensitively with a lower limit of detection. Further development of the method may provide a more convenient, highly sensitive, and effective detection for BPA in complex samples.

## Figures and Tables

**Figure 1 micromachines-11-00041-f001:**
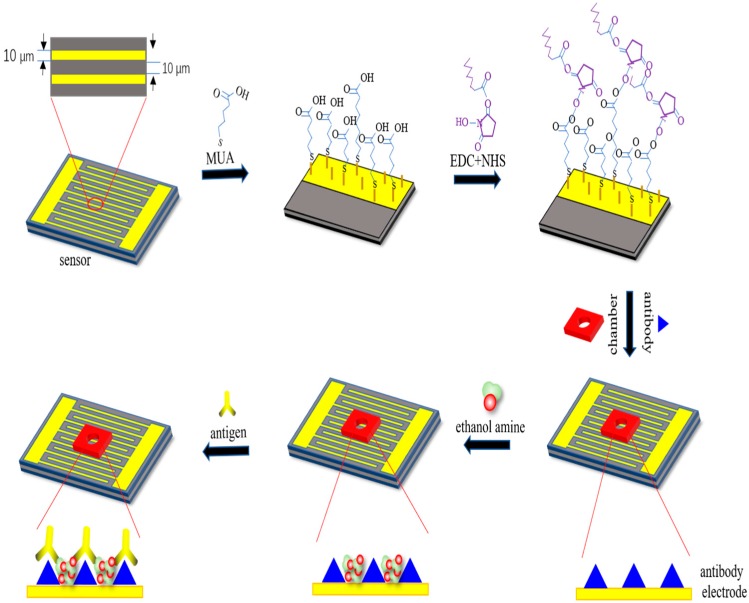
Representation of surface modification techniques on the interdigital electrodes’ surface for the detection of bisphenol A (BPA).

**Figure 2 micromachines-11-00041-f002:**
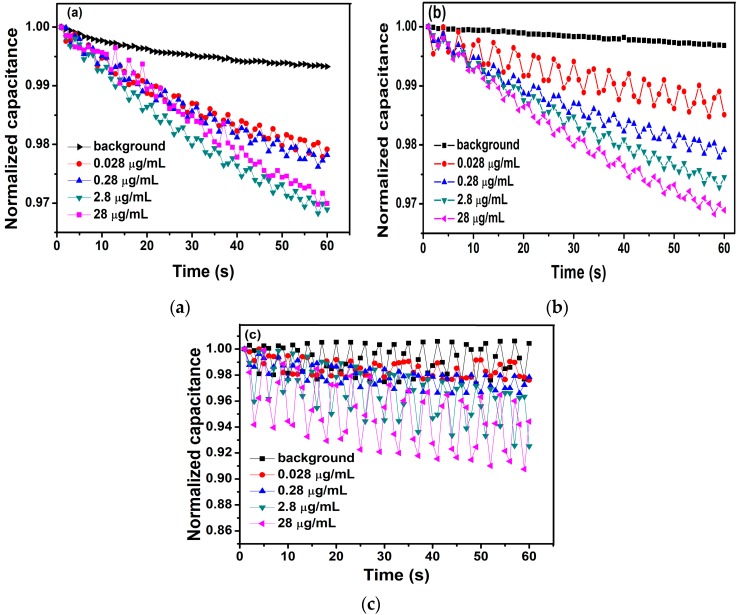
The relationship between the normalized capacitance and different concentrations of BPA with a 10-kHz AC signal of different voltages ((**a**) 100 mV, (**b**) 600 mV, and (**c**) 1.1 V) applied to the interdigital electrodes.

**Figure 3 micromachines-11-00041-f003:**
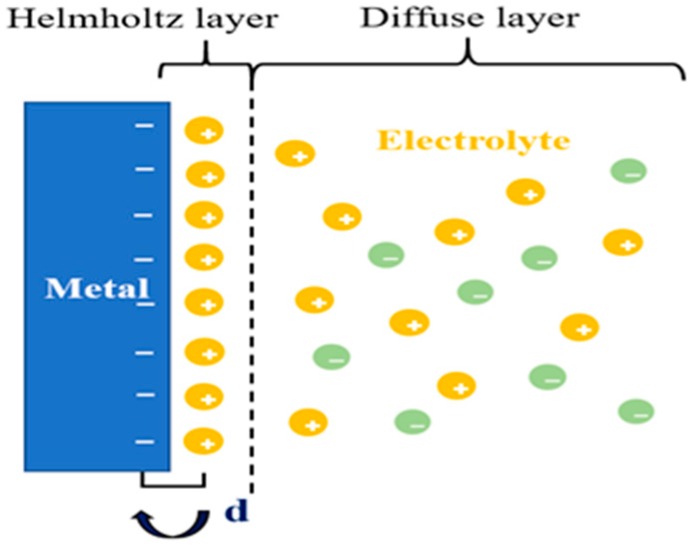
The double layers show the opposite charges equally distribute on both sides of the interface.

**Figure 4 micromachines-11-00041-f004:**
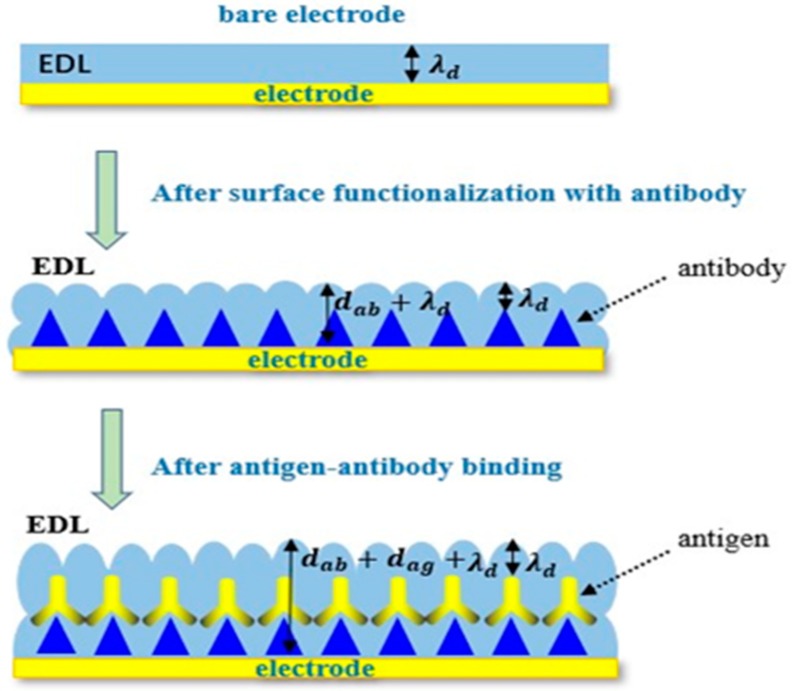
Changes on the electrode surface due to the specific binding of antigens to antibodies.

**Figure 5 micromachines-11-00041-f005:**
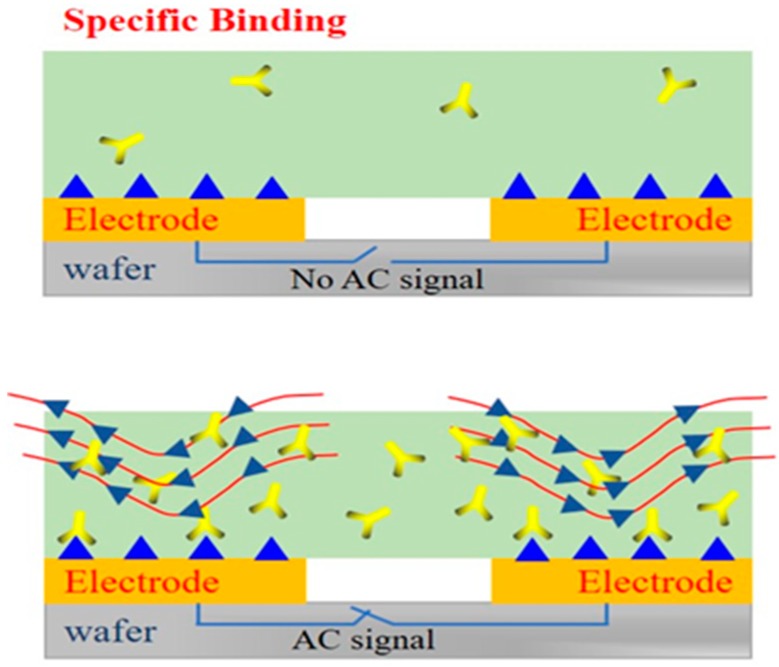
The performances of different detection environments on electrodes.

**Figure 6 micromachines-11-00041-f006:**
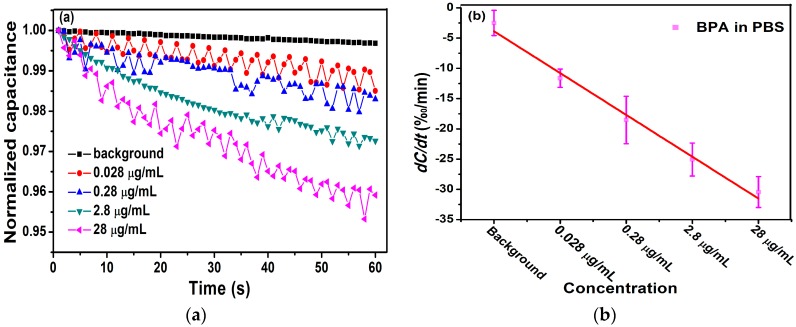
(**a**) Detection of different concentrations of antigen with 5.3 μg/L antibody. (**b**) The change rate of capacitance as a function of BPA concentrations in 0.01 × PBS.

**Figure 7 micromachines-11-00041-f007:**
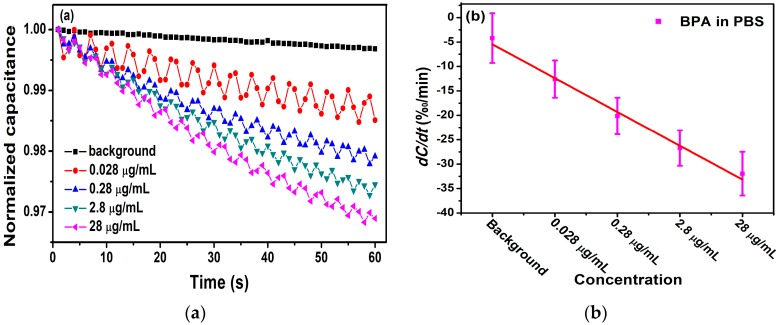
(**a**) Detection of different concentrations of antigen with 5.3 μg/mL of antibody. (**b**) The change rate of capacitance as a function of BPA concentrations in 0.01 × PBS.

**Table 1 micromachines-11-00041-t001:** Limit of detection (LOD) comparison of different methods.

Method	LOD	Reference
Carbon nanohorns/Nafion	1.8 $\times 10^{-6}$ mol/L	Yilin Li (2014)
SDS-(bupy)P$F_{6}$/CPE	3.02 $\times 10^{-7}$ mol/L	Zhang Yanmei (2012)
Graphene modified SPCE	4.99 $\times 10^{-7}$ mol/L	Ling Zhou (2014)
Strata^®^ C18-E cartridge cleanup with detection by liquid chromatography coupled	5.21 $\times 10^{-9}$ mol/L	Pasquale Gallo (2019)
Nanoparticles-based fluorescence immunoassay	8.7 $\times 10^{-11}$ mol/L	Wei Sheng (2018)
Surface plasmon resonance (SPR) biosensor	2.28 $\times 10^{-11}$ mol/L	Xue C S (2019)
Gold nanoparticle-based colorimetric aptasensor	4.38 $\times 10^{-12}$ mol/L	Eun-Hee Lee (2019)
Electroactive label-based aptamer	3.8 $\times 10^{-10}$ mol/L	Maryam Nazari (2019)
A tyrosinase-based biosensor	7.7 $\times 10^{-8}$ mol/L	Filomeno A.D. Inroga (2019)
Electrochemical enzyme biosensor	3.18 $\times 10^{-9}$ mol/L	Yang Liu (2019)
A signal-enhanced lateral flow strip biosensor	4.38 $\times 10^{-8}$ mol/L	Xiayu Peng (2017)
DNA-functionalized graphene field effect transistors integrated in microfluidic systems.	4.38 $\times 10^{-9}$ mol/L	Liu S (2018)
Self-assembly technology and AC electrokinetics effect	1.22 $\times 10^{-7}$ mol/L	This work

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
