# Peer review of "Rapid and Sensitive Detection of Bisphenol A Based on Self-Assembly"

_micromachines, 2019, doi:10.3390/mi11010041_

Round 1
Reviewer 1 Report
The authors must clarify precisely the differences between this work and the previous work of the authors:
[1] Lin, X., Cheng, C., Terry, P., Chen, J., Cui, H., Wu, J. Rapid and sensitive detection of bisphenol a from serum matrix (2017) Biosensors and Bioelectronics, 91, pp. 104-109.
The introduction shows the following: “In this work, a rapid, highly sensitive BPA biosensor based on self-assembly technology and AC electrokinetics (ACEK) method was developed, which could accomplish the rapid and sensitive detection of BPA.” However, this method was already used in [1]. All the differences with respect to the previous article must be specified.
Similarly, the novel aspects of the proposed technology must be explicitly detailed in the methodology and detailed and clarified regarding [1].
In [1] the following is specified: “A BPA aptamer specific to BPA with 5’ thiol modifier C6 SH (Fisher Scientific, PA) was immobilized ”. However, this work does not specify details and characteristics of the Antibody. The differences between the two studies must be specified. Antibody specification and characteristics must be made.
The introduction does not review the state of the art in relation to other systems proposed by other authors for the measurement of BISPHENL A. Authors should conduct a thorough literature review that highlights the novelty of the work.
The results obtained in [1] are expressed in fM, and the results of the present work in ug / liter. Current results should be compared with those obtained in [1] and added to table 1.
In the introduction and in the comparative, the advantages and novelties of the proposed System should be highlighted compared to recent publications on the subject. The advantages with respect to the references in Table 1 are clear, but not with respect to recent publications:
Wu, L., Gao, J., Lu, X., Huang, C., Dhanjai, Chen, J.Graphdiyne: A new promising member of 2D all-carbon nanomaterial as robust electrochemical enzyme biosensor platform(2020) Carbon, 156, pp. 568-575.
Mayedwa, N., Ajayi, R.F., Mongwaketsi, N., Matinise, N., Mulaudzi-Masuku, T., Hendricks, K., Maaza, M.Development of a Novel Tyrosinase Amperometric Biosensor Based on Tin Nanoparticles for the Detection of Bisphenol A (4.4-Isopropylidenediphenol) in Water(2019) Journal of Physics: Conference Series, 1310 (1).
Lou, C., Jing, T., Tian, J., Zheng, Y., Zhang, J., Dong, M., Wang, C., Hou, C., Fan, J., Guo, Z.3-Dimensional graphene/Cu/Fe3O4 composites: Immobilized laccase electrodes for detecting bisphenol A(2019) Journal of Materials Research, 34 (17), pp. 2964-2975.
Xue, C.S., Erika, G., JiÅ™í, H.Surface plasmon resonance biosensor for the ultrasensitive detection of bisphenol A(2019) Analytical and Bioanalytical Chemistry, 411 (22), pp. 5655-5658.
Nazari, M., Kashanian, S., Rafipour, R., Omidfar, K.Biosensor design using an electroactive label-based aptamer to detect bisphenol A in serum samples(2019) Journal of Biosciences, 44 (4) .
Jalalvand, A.R., Haseli, A., Farzadfar, F., Goicoechea, H.C.Fabrication of a novel biosensor for biosensing of bisphenol A and detection of its damage to DNA(2019) Talanta, 201, pp. 350-357.
Lee, E.-H., Lee, S.K., Kim, M.J., Lee, S.-W.Simple and rapid detection of bisphenol A using a gold nanoparticle-based colorimetric aptasensor(2019) Food Chemistry, 287, pp. 205-213.
Inroga, F.A.D., Rocha, M.O., Lavayen, V., Arguello, J.Development of a tyrosinase-based biosensor for bisphenol A detection using gold leaf–like microstructures(2019) Journal of Solid State Electrochemistry, 23 (6), pp. 1659-1666.
Allsop, T.D.P., Neal, R., Wang, C., Nagel, D.A., Hine, A.V., Culverhouse, P., Ania Castañón, J.D., Webb, D.J., Scarano, S., Minunni, M.An ultra-sensitive aptasensor on optical fibre for the direct detection of bisphenol A(2019) Biosensors and Bioelectronics, 135, pp. 102-110.
Liu, Y., Yao, L., He, L., Liu, N., Piao, Y.Electrochemical enzyme biosensor bearing biochar nanoparticle as signal enhancer for bisphenol a detection in water(2019) Sensors (Switzerland), 19 (7).
UludaÄŸ, Y., Ölçer, Z., DoÄŸan, C., Muhammad, T., AltintaÅŸ, Z.Rapid and on-site electrochemical detection of bisphenol A and arsenic in drinking water using a novel electrode array(2019) Turkish Journal of Chemistry, 43 (2), pp. 612-623.
Wang, C.-Y., Zeng, Y., Shen, A.-G., Hu, J.-M.A highly sensitive SERS probe for bisphenol A detection based on functionalized Au@Ag nanoparticles(2018) Analytical Methods, 10 (47), pp. 5622-5628.
Peng, X., Kang, L., Pang, F., Li, H., Luo, R., Luo, X., Sun, F.A signal-enhanced lateral flow strip biosensor for ultrasensitive and on-site detection of bisphenol A(2018) Food and Agricultural Immunology, 29 (1), pp. 216-227.
Liu, S., Fu, Y., Xiong, C., Liu, Z., Zheng, L., Yan, F.Detection of Bisphenol A Using DNA-Functionalized Graphene Field Effect Transistors Integrated in Microfluidic Systems(2018) ACS Applied Materials and Interfaces, 10 (28), pp. 23522-23528.
Ben Messaoud, N., Ghica, M.E., Dridi, C., Ben Ali, M., Brett, C.M.A.A novel amperometric enzyme inhibition biosensor based on xanthine oxidase immobilised onto glassy carbon electrodes for bisphenol A determination(2018) Talanta, 184, pp. 388-393.
Author Response
Thanks for your professional comments and suggestions. We have responsed to your all comments in the attachment. Please see the attachment.

Reviewer 2 Report
This manuscript shows experimental results of a sensitive biosensor for bisphenol A (BPA) based on impedance spectroscopy exploiting surface functionalization in interdigitated electrodes using MUA. Since BPA is a toxic substance that can cause serious diseases, including diabetes and heart disease, the development of sensitive and low-cost BPA sensors can have very good societal impacts.
The authors need to add some clarifications to the paper to address a few questions and concerns that are listed below:
The authors indicated that the use of the proposed approach with a 10 kHz AC signal of 600 mV enhanced the performance of the method due to AC electrokinetics (ACEK). However, the authors did not provide any results to justify the impact of ACEK in the results obtained. The authors should justify the contribution of ACEK in the performance of the sensor through the application of the same 10 kHz AC signal with different voltages, such as 300 mV and 1.2 V. The authors need to indicate if the voltage used in the experiments is peak, peal-to-peak, or rms.
Was there a process used to determine the frequency 10 kHz used in the experiments? If so, that should be included in the paper.
The authors included the following sentence on page 93: “Before detecting antigen concentrations, each sensor should detect the background solution.” This means that the operator needs to provide the solution under test without the antigen. Doesn’t this requirement undermine the practical application of this sensor? since one need to have access to a solution without the analyte before carrying out the sensing? Do the commercially available sensors used to detect BPA also have this requirement?
Based on the schematic of the protocol for the sensor preparation shown in Fig. 2.3, the sensor needs to be cleaned and the surface functionalized with antibodies again after each sensing procedure. Do the authors foresee any way to reuse the sensor multiple times without requiring surface functionalization after each use? If this is the case, the proposed interdigitated electrode biosensor would have to be disposable due to the lack of facilities to carry out the functionalization at the testing site. How does that compare with other BPA sensors?
On the Line 123, the authors wrote “electrolyte will contract with the surface of microelectrode in the Fig 1.”. The authors need to clarify this statement, especially the word “contract”.
The authors need to clarify the statements in the lines 143-145. How will this measurement overcome the electrode imperfections and experimental differences?
On the lines 171-173, the authors need to clarify the sentence: “The change rate of normalized capacitance curves was found by least square linear fitting, which provided a quantitative index of electrode’s self-assembly.” Essentially, what is the correlation between the sensitivity of the biosensor and the self-assembly?
On the line 209, does the expression "better binding" used by the authors mean "larger number of antibody-antigen binding"? This issue needs to be clarified.
The expression in the lines 210 and 211: “the smaller impedance of the self-assembly interdigital electrode sensors had” needs to be fleshed out to facilitate the understanding.
The authors use ul, uL, μL to represent microliter throughout the text an in the keys of Fig. 5(a). The authors should use a single notation to represent microliter. The most suitable notation for microliter is μL, which the authors should use throughout the paper.
The authors need to introduce the acronyms MUA and EDC used in the paper.
Author Response

(The authors gave the same response as above.)

Reviewer 3 Report
The authors have presented the investigation results on newly developed BPA biosensors based on self-assembly and AC electrokinetics capacitive sensing methods. The developed sensors exhibited higher sensitivity in comparison with other results reported by current detection methods. In the present form, the paper is well-organized, some English corrections are needed. Nevertheless, it can be accepted based on the presented merit.
Author Response
Thanks for your professional comments and suggestions. We have done some English corrections in the manuscript.
Round 2
Reviewer 1 Report
NEW COMMENTS ABOUT THE PREVIOUS COMMENTS MUST BE ADDRESSED FOR THE ARTICLE TO BE SUITABLE FOR PUBLICATION (SEE NEW COMMENTS):
The authors must clarify precisely the differences between this work and the previous work of the authors:
[1] Lin, X., Cheng, C., Terry, P., Chen, J., Cui, H., Wu, J. Rapid and sensitive detection of bisphenol a from serum matrix (2017) Biosensors and Bioelectronics, 91, pp. 104-109.
Point 1: The introduction shows the following: “In this work, a rapid, highly sensitive BPA biosensor based on self-assembly technology and AC electrokinetics (ACEK) method was developed, which could accomplish the rapid and sensitive detection of BPA.” However, this method was already used in [1]. All the differences with respect to the previous article must be specified.
Response 1: Thanks for your professional comments. Firstly, in this work, we used the BPA antibody to achieve the detection of BPA. While an aptamer probe was used to detect BPA in [1]. Secondly, we used the self-assembly technology to form self-assembled monolayer with the mercaptan containing alkyl chains immobilized on the gold electrodes via Au-S bonds. Then N-hydroxysuccinimide (NHS) and 1-(3- dimethylaminopropyl)-3- ethyl carbon diimide hydrochloride (EDC) were used as the crosslinking agent to assist in the formation of amide bonds between the carboxyl group of the self-assembled monolayer and the amino group of the BPA antibody. With these processes, the BPA antibody was immobilized onto the electrodes stably. While a BPA aptamer specific to BPA with 5’ thiol modifier C6 SH (Fisher Scientific, PA) was directly immobilized onto the electrodes in [1]. Thirdly, in this work, the interdigital microelectrodes had interdigitated arrays with widths of 10μm separated by 10μm gaps. However, the interdigital microelectrodes had interdigitated arrays with widths of 6μm separated by 6μm gaps in [1]. Lastly, in this work, a 10 kHz AC signal was applied to the microelectrodes. But a 20 kHz AC signal was applied to the microelectrodes in [1].
NEW COMMENT: The differences between both works must be mentioned explicitly in the manscrit. Please include the previous explanation in the work. Please, indicate in the answer in which exact lines the modifications have been made.
Similarly, the novel aspects of the proposed technology must be explicitly detailed in the methodology and detailed and clarified regarding [1].
Point 2: In [1] the following is specified: “A BPA aptamer specific to BPA with 5’ thiol modifier C6 SH (Fisher Scientific, PA) was immobilized”. However, this work does not specify details and characteristics of the Antibody. The differences between the two studies must be specified. Antibody specification and characteristics must be made.
Response 2: Thanks for reminding us. Compare to the work in [1], in this work, we used the self-assembly technology to form self-assembled monolayer with the mercaptan containing alkyl chains immobilized on the gold electrodes via Au-S bonds. Then N-hydroxysuccinimide (NHS) and 1-(3- dimethylaminopropyl)-3- ethyl carbon diimide hydrochloride (EDC) were used as the crosslinking agent to assist in the formation of amide bonds between the carboxyl group of the self-assembled monolayer and the amino group of the BPA antibody. With these processes, the BPA antibody was immobilized onto the electrodes stably.
NEW COMMENT: Please include this comment into the manuscript. Please, indicate in the answer in which exact lines the modifications have been made.
Point 3: The introduction does not review the state of the art in relation to other systems proposed by other authors for the measurement of BISPHENL A. Authors should conduct a thorough literature review that highlights the novelty of the work.
Response 3: Thanks for your professional advice. We have reviewed the state of the art in relation to other systems proposed by other authors for the measurement of BPA in the introduction of the manuscript.
NEW COMMENT: OK
Point 4: The results obtained in [1] are expressed in fM, and the results of the present work in ug / liter. Current results should be compared with those obtained in [1] and added to table 1.
Response 4: Thanks for reminding us. The present work could successfully detect BPA at nanomolar (nM) levels which was higher than the results in [1], which could successfully detect BPA at femto molar (fM) levels. Though the results of the present work were not better than the results in [1], we have used a novel method (self-assembly technology) to achieve the detection of BPA. At the same time, in order to acquire the better results, we are improving the conditions, processes and materials of the experiments. And the novel method has the tremendous potential to detect BPA more sensitively with lower limit of detection.
NEW COMMENT: Please include this comment into the manuscript. Please, indicate in the answer in which exact lines the modifications have been made.
Point 5: In the introduction and in the comparative, the advantages and novelties of the proposed System should be highlighted compared to recent publications on the subject. The advantages with respect to the references in Table 1 are clear, but not with respect to recent publications:
Response 5: Thanks for your professional advice. Compare to the following publications on the subject, the results of the present work may not be better than them. And most of the recent publications have reported the electrochemical method to achieve the detection of BPA. For example, Maryam et al. have used an electroactive label-based aptamer to detect bisphenol A in serum samples with the linear range of 0.2–2 nM. Inroga et al. have used gold leaf–like microstructures to develop a tyrosinase-based biosensor for bisphenol A detection with the limit of detect of 0.077μM. However, in this work, we have demonstrated a novel method to detect BPA based on self-assembly technology and AC electrokinetics effect. And there is nearly no report to detect BPA with self-assembly technology and AC electrokinetics effect. So it has the tremendous potential to detect BPA more sensitively with lower limit of detection in the near future.
NEW COMMENT: Please include this comment into the manuscript. Please, indicate in the answer in which exact lines the modifications have been made. Please include a LARGE COMPARTATIVE TABLE of the quantitative results obtained with respect to recent publications.
Author Response

(The authors gave the same response as above.)

Reviewer 2 Report
Comments provided in the attached file.

Author Response

(The authors gave the same response as above.)

Round 3
Reviewer 1 Report
All comments have been adequately answered and the article is suitable for publication.